# GPS Suitability for Physical Frailty Assessment

**DOI:** 10.3390/s24144588

**Published:** 2024-07-15

**Authors:** Pablo Boronat, Miguel Pérez-Francisco, Arturo Gascó-Compte, Miguel Pardo-Navarro, Oscar Belmonte-Fernández

**Affiliations:** 1Computer Languages and Systems Department, Universitat Jaume I (UJI), Av. Sos Baynat s/n, 12071 Castellón, Spain; boronat@uji.es (P.B.); agasco@uji.es (A.G.-C.); mipardo@uji.es (M.P.-N.); belfern@uji.es (O.B.-F.); 2Computer Science and Engineering Department, Universitat Jaume I (UJI), Av. Sos Baynat s/n, 12071 Castellón, Spain

**Keywords:** GPS monitoring, frailty assessment, gait speed, daily monitoring

## Abstract

The ageing of the population needs the automation of patient monitoring. The objective of this is twofold: to improve care and reduce costs. Frailty, as a state of increased vulnerability resulting from several diseases, can be seen as a pandemic for older people. One of the most common detection tests is gait speed. This article compares the gait speed measured outdoors using smartphones with that measured using manual tests conducted in medical centres. In the experiments, the walking speed was measured over a straight path of 80 m. Additionally, the speed was measured over 2.4 m in the middle of the path, given that this is the minimum distance used in medical frailty tests. To eliminate external factors, the participants were healthy individuals, the weather was good, and the path was flat and free of obstacles. The results obtained are promising. The measurements taken with common smartphones over a straight path of 80 m are within the same order of error as those observed in the manual tests conducted by practitioners.

## 1. Introduction

As the population in many developed countries is becoming older, there is a crucial need to reduce the care costs for older people. In this context, frailty assessment is an important aspect, given that early detection could accelerate the decision to apply less expensive therapies to prevent physical deterioration [1].

Frailty tests can be conducted on a patient during medical visits. The low frequency of medical visits makes it difficult to detect pre-frailty symptoms at a stage in which soft therapies could have been applied. Fortunately, continuous activity monitoring can now be performed with common smartphones or smartwatches.

An important research effort has been devoted to the continuous detection or monitoring of diseases such as Alzheimer’s, pulmonary, or arterial diseases, and more generally, frailty, using wearable devices. However, data extracted from real-life monitoring must be validated against medical test indicators.

Walking speed is one of the most commonly used tests to assess a patient’s frailty by medical practitioners. In particular, the walking speed is usually measured in a medical centre and the test is limited to a straight path of few meters in a room [2]. Consequently, these tests are applied infrequently over time and, in addition, they represent a tedious and time-consuming task for practitioners.

The goal of our study is to validate the measurement of gait speed outdoors using the GPS unit of commonly used smartphones as the only instrumentation. To do this, the speed obtained with the smartphone is compared with a photo-detection installation, which is considered the ground truth. Moreover, the accuracy of these measurements is compared with that of the manual tests conducted in medical centres. The main contribution of this article is to verify that smartphone measurements of gait speed over distances of tens of meters are as accurate as indoor manual tests of a few meters.

In the experiments, we obtained differences lower than 4.7% between the indoor manual speed tests for a distance of 2.4 m and GPS measurements for a path of 80 m. Comparing both methods (manual and GPS measurements) for short distances of a few meters, the differences are lower than 8.7%, mainly due to the greater influence of GPS error.

If walking speed could be assessed continuously during the real-life activity of patients, frailty could be evaluated more frequently. Additionally, the test would be more representative of the patient’s state as longer paths could be measured.

The rest of this paper is organised as follows. Section 2 is devoted to introducing the state of the art. The context of the experiments is described in Section 3. In Section 4, the results obtained are presented and analysed. Finally, the main conclusions, future work and discussion are included in Section 5 and Section 6.

## 2. Related Work

GPS tracking is a part of our daily lives. It is extensively used in driving, working, and engaging in sports. With its widespread deployment, numerous academic studies have explored the application of GPS and other technological devices in monitoring natural behaviour. These investigations typically pursue two objectives: the frequent and representative monitoring of patients, and reducing the cost of medical tests and therapies. The expected results include more detailed data (both in terms of quantity and quality) to track patients’ progression, alongside cost reductions for health services [1,3,4].

A general review of the use of GPS monitoring to study active living is shown in [5]. This review highlights the significance of GPS monitoring for tracking patient activity. It emphasises not only physical activity but also social behaviour, which is considered crucial in the evolution of frailty, as it has been demonstrated in studies such as [6].

The literature related to monitoring natural behaviour for medical tracking can be roughly classified into two groups. In the first group, in which the present paper is included, the studies are devoted to the validation of new devices and procedures that could be used for “continuously” monitoring, by comparing them with established clinic tests [3,7,8,9]. The second group comprises publications dealing with the analysis of data extracted from natural behaviour monitoring in order to filter or correlate the representative data for the survey of frailty (or other diseases) evolution [1,4,10,11,12]. Concerning the second group, a review on frailty prediction methods can be found in [13].

Early works, such as [7], previewed the use of GPS technology to measure and characterise the gait speed in outdoor scenarios. The authors used expensive GPS receivers and applied techniques such as differential GPS to obtain high precision. The technical complexity makes it difficult for these methods to be widely adopted. Since then, GPS technology has been improved, and it is massively used nowadays. Using commonly available smartphones, we can achieve errors of less than 0.12 m/s for gait speed. This precision is sufficient for the purpose of our work, especially over walking distances of tens of meters. In any case, the authors will state the biases in measuring gait speed, and they will point out the main problems, which are still to be assessed further in Section 6.

In 2008, Le Faucher et al., in the context of peripheral arterial diseases, compared the standard way to measure the maximal walking distance using a treadmill with other methods such as questionnaires, the 6 min walk test, and GPS tracking of predefined outdoor walking [3]. The authors found that the most correlated method to the standard is GPS monitoring, and they showed that the results could be more representative than those from the treadmill test. However, the authors pointed out that the effect of external factors such as the weather, ground surface, or the slope of the path is still to be studied. This work is similar to ours, but we focus on the 4 m speed test as part of physical frailty detection.

Later on, apparent contradictory articles about the maximal walking distance test for peripheral arterial patients were published [14,15]. Letters to the editor of the Circulation journal concerning this controversial were sent [16,17]. In the case of the 4 m walking speed test, what is relevant is to reflect the natural walking speed rather than the patient’s endurance.

In [8], a validation of natural walk monitoring with respect to clinical assessments for slow-walking subjects is presented. This study shares our long-term goal; however, they use a waist-worn inertial sensor to detect the steps. Their results probably can be extrapolated to the sensors already present in smartphones and smartwatches (e.g., accelerometers and gyroscopes). However, for outdoor monitoring, the gait speed requires the use of GPS data to calculate the distance travelled over time.

A validation of smartphones for medical measurements is presented in [9]. In the article, the authors compare the gait speed obtained in the 6 min walk test in indoor environments using medical instruments, smartphones (with an adapted firmware which was developed in a previous work), and commonly used applications for fitness tracking. In the article, the authors also compare six regression models based on machine learning. Phone sensors such as accelerometers are used to deduce the gait speed and distance travelled. These results, and related studies, could be combined with our approach for outdoor or mixed monitoring to provide a wider spectrum of patient data.

There are some remarkable works to detect or track the evolution of frailty in older adults. In [10], three methods to define the activity or mobility of older adults from GPS monitoring are used. Another work devoted to monitoring physical activity is presented in [1]. In this work, an algorithm processing GPS data allows the authors to differentiate home activity from outdoor activity. In [18], GPS tracking is used to study the activity spaces of seniors in order to deduce aspects related to health, including social traits.

Similar studies on the use of GPS monitoring to study dementia evolution through outdoor activity using GPS devices are referenced in [4,12]. In [4], 12 mobility indicators are proposed. These indicators do not just quantify spatial characteristics but also temporal ones in order to capture outdoor complex behaviour analysis. In this work, the authors add a semantic dimension to the mobility by looking for the type of end points of trips and then trying to infer the type of activity the patient is performing. In [12], outdoor displacements are dissected also to obtain complexity indicators of the trajectories. In this case, the trajectories are discretised and treated as graphs. A reduction in the complexity of the daily trips should point to Alzheimer’s disease symptoms. This kind of study, while being fundamental for the evolution of our project, is beyond the scope of the objective of the present paper. The goal of our work is to compare the error in the gait speed measured using common smartphones outdoors with manual indoor tests in order to know if it could complement the medical frailty tests.

In [11], the authors automate data collection in a non-intrusive manner during activities of daily living for ecological frailty assessment using wearable devices. Then, the frailty syndrome is classified into three classes applying the Fried test using machine learning techniques. In this article, a microservices-based architecture is proposed, making possible the extension of the system with different frailty model microservices or including different sensors.

Despite the existing relevant related articles, none have specifically determined whether the gait speed measured with smartphones or similar wearable devices is representative of the walking speed tests applied in medical frailty diagnosis.

## 3. Experimental Setup

A flat and straight path of 80 m, shown in Figure 1, was used for the tests. This path is surrounded by buildings and trees inside the University Jaume I campus, being a realistic urban scenario regarding the quality of the GPS signal. This scenario was chosen to study the viability of smartphone monitoring as a complement to medical assessments for checking frailty evolution. In this preliminary work, we have tried to reduce the impact of external conditions.

A schema showing the geometry of the experiments is presented in Figure 2. Along the experimental path, four photo-diodes were installed to detect the presence of the participants as the ground truth to calculate the walking speed (PD1 to PD4 in the figure). The photo-diodes were placed at distances 0, 39, 41.4 and 80 m from the starting line. Those placed at 39 and 41.4 m (PD2 and PD3 in the figure) were connected to an Arduino microcontroller, which sent UDP messages to a computer server through an Ethernet network. Photo-diodes placed at distances 0 and 80 m (PD1 and PD4) were connected to respective ESP32 microcontrollers, which, in turn, sent UDP messages to the computer server using a dedicated Wi-Fi access point. Communication time between the microcontrollers and the server computer was negligible with respect to the phenomenon to be measured: UDP messages were sent every few seconds and communication time was less than 50 ms. Tracking data extracted by the smartphone were sent offline (after the walk). During the experiments, no message losses were observed.

To ensure a stable speed before measurements, the participants in the tests began to walk 4 m before the starting point and continued to walk 4 m past the arrival point (placed at 80 m). Additionally, the application developed for the experiments was initiated before the walking started to ensure that the GPS unit was ready to provide geolocations with sufficient accuracy. The participants were asked to touch a button in the phone application when they crossed the initial and ending lines of the path. This action started (stopped) data recording.

The validation of the gait speed obtained with the GPS unit of common smartphones could be performed with any fitness application. Nevertheless, for our experiments, we preferred an application developed ad hoc because it allowed us to easily control the sampling frequency of the sensors and how the files of each walk were sent.

The photo-diodes separated 2.4 m apart were used to detect the speed at the middle of the walking path. In the case of the application, the speed in the central part was measured using three consecutive samples from the GPS (two seconds of walking, given that the GPS updates the position at a minimum interval of one second).

The conditions of the experiments were chosen to eliminate all external factors. A total of 8 healthy adults participated in the experiments, 3 female and 5 male, aged between 24 and 60. Each participant walked 5 times, providing a total of 40 walks. Regarding the rest of the conditions, the path was flat and straight. All tests were conducted under the same weather conditions: a sunny day with moderate temperature (20 °C) and no wind. The same phone was used in the experiments (Redmi Note 9 pro with Android v11, Xiaomi, Beijing, China). Also, all participants held the phone in their hand.

The selection of the participants in the experiments was performed to cover a wide range of ages, including both males and females, all of whom were in good health. Given that frail patients usually walk slowly, even better results would be expected. Another aspect to mention is the reduced number of participants. Taking into account the stability of the results and the maturity of the technology being applied, we consider that it is significant for the phenomenon under study. Figure 3 shows the average difference between gait speed obtained by photo-detection and with the smartphone (i.e., the average error) as a function of the number of walks along the 80 m path. The average difference oscillates between 0.035 and 0.039 m/s after 25 walks and then stabilises at around 0.039 m/s.

## 4. Results

The results of our tests are presented below, addressing various questions.

Figure 4 shows the average gait speed and the standard deviation for each participant. The data in this figure correspond to the speed over 80 m measured by photo-detection. In this figure, it can be seen that the average speed of each participant is quite significant, as the standard deviations are lower than 0.07 m/s, which is 5% of the average speed for the participant with the highest deviation. In addition, as we have tried to minimize the external factors in the experimental setup, no outlier measures have been observed. In the results, all the measures have been considered representative.

### 4.1. Question 1—Is the Speed Measured with the GPS Accurate Enough for the 80 m Walk?

The application developed for the tests records several phone sensors and sends all data to a server at the end of the walk. Then, with the registered positions and times, the speed of walking is computed.

The walking speed measured with the smartphone is compared with the reference speed obtained with the photo-diodes placed at the two ends of the 80 m path. Figure 5 shows the relation between the speeds obtained for all the walks with the smartphone and the photo-diodes, including the linear regression line. The mean absolute error is 0.04 m/s, and the maximum difference is 0.12 m/s.

In Figure 5, the value of the slope for the linear regression is 1.0723 and the *p*-value of the independent variable (the speed measured with the photo-diodes) is almost zero ($2.691\times 10^{-20}$), showing the statistical significance between both methods. The intercept’s value is close to 0 (−0.0850), and its *p*-value (0.323) is significantly larger than 0.05, which implies a high probability of the actual intercept’s value being zero. Furthermore, the value of the coefficient of determination $R^{2}$ is 0.890, which also confirms the linear relationship between the two variables.

### 4.2. Question 2—Is the Speed Measured by the GPS over the Distance of a Short Path Accurate Enough?

In this case, the speed measured by the application in a sub-path in the middle of the path is compared with the reference speed measured by photo-diodes separated by 2.4 m (placed at 39 and 41.4 m of the path). This result is interesting because of the fact that medical tests are reduced to a walk distance of few meters (at least 2.4 m, as indicated in [2]) in medical centres.

Using the smartphone, three consecutive locations were taken, as the smartphone’s GPS is limited to one measurement per second. Thus, the distance measured corresponds to two seconds. For this short distance, bigger differences between the two methods were expected, given the weight of the GPS error on the measures.

The relation between the speeds measured by photo-detection and with the smartphone, including the linear regression line, for a distance of 2.4 m can be seen in Figure 6. In this case, the mean absolute error is 0.098 m/s, and the differences observed are lower than 0.26 m/s. The value of the slope for the linear regression is 0.8768 and the *p*-value of the independent variable (the speeds measured by photo-detection) is almost zero ($5.038\times 10^{-8}$), which confirms the statistical significance between both methods. The intercept’s value is 0.218, and its *p*-value (0.231) is significantly larger than 0.05, which implies a high probability of the actual intercept’s value being zero. Furthermore, the value of the coefficient of determination $R^{2}$ is 0.537, which means that there is a linear relationship between the two variables.

Even though the differences are bigger than that observed for the distance of 80 m, the regression shown in Figure 6 validates the hypothesis according to which the speed measured with the smartphone could be used in the walking speed test applied by practitioners to check the patient’s frailty evolution.

### 4.3. Question 3—Is the Walking Speed over the distance of a Short Sub-Path Representative of the Speed over the Distance of a Long Path?

Walking speed tests of long paths (in the order of hundreds of meters) are more representative of the natural walking speed. Consequently, these tests are more reliable to check the frailty state of older adults. A patient could have an acceptable gait speed for a distance around 3 m, but they could be incapable of maintaining this speed when walking naturally.

In any case, given that indoor medical speed tests are reduced to a distance between 2.4 and 10 m, it is worth comparing the speed over a distance of a short path or 2.4 m in the middle of the 80 m path with the speed over the whole distance of 80 m. This would prove that the gait speed for healthy people in a controlled ambience (i.e., weather, ground slope, …) is stable.

Figure 7 shows the relation between the speed measured with the photo-diodes over the distances of 80 m and 2.4 m. As previously said, the photo-diodes separated 2.4 m apart are in the central part of the 80 m path. The difference between the two measurements is lower than 0.09 m/s.

In Figure 7, the value of the slope for the linear regression is 1.0365, and the *p*-value of the independent variable (the speed measured with the photo-diodes for the 80 m tests) is almost zero ($2.662\times 10^{-16}$), showing the statistical significance between both methods. The intercept’s value is −0.0828, and its *p*-value (0.447) is significantly larger than 0.05, which implies a high probability of the actual intercept’s value being zero. Furthermore, the value of the coefficient of determination $R^{2}$ is 0.824, which again confirms the linear relationship between the two variables.

### 4.4. Question 4—What Is the Importance of the GPS Error for the Speeds Measured over Long and Short Paths?

Figure 7 shows that the participants walk at a fairly constant speed along an 80 m straight distance. Even if this can be deduced from previous figures, Figure 8 presents the relation between the speeds measured with the smartphone’s GPS unit over the distance of 80 m and the intermediate speeds between 2 s of walking.

Figure 8 mainly points out the weight of the GPS error on the speed over the distances of a short and a long path. In this figure, the value of the slope for the linear regression is 0.967, which proves that the data are closely related. The *p*-value of the independent variable (the speed measured with the GPS over two seconds) is almost zero ($3.288\times 10^{-10}$). The intercept’s value is 0.0457, and its *p*-value (0.782) is significantly larger than 0.05, meaning that the intercept is probably zero. Also, the linear relationship between the two variables is confirmed by the coefficient of determination, $R^{2}=0.651$.

### 4.5. Question 5—How Good Are GPS Measurements with Respect to Manual Speed Tests?

We have performed manual indoor gait speed tests with a distance of 2.4 m using a chronometer, as health practitioners tend to do. Again, these measures are compared with photo-detection, which is considered the reference ground truth. In these experiments, four practitioners measured 204 walks distributed over several sessions.

Figure 9 shows the relation between the photo-detection and manual measurements over 2.4 m. The line in the figure is the linear regression, with a slope of 0.947 and an intercept of 0.0670.

The MAE (mean absolute error) calculated by the practitioners and the speed measured by GPS can be seen in Figure 10. This figure shows that the speeds measured by GPS for a distance of 80 are of the same level of error as the manual tests. However, for short distances (three GPS positions for two seconds), the error of the measurements with the GPS is greater (0.098 m/s).

## 5. Discussion

In this article, we present a study that considers using the GPS unit of a smartphone to measure the gait speed in order to continuously monitor the frailty state of patients.

In the experiments, the GPS measurements are compared with those taken by photo-detection, which is considered the ground truth. Throughout these tests, a very high correlation is observed between the photo-detection measurements and the speed obtained with the smartphone. The maximum, minimum and average observed differences between the GPS and photo-detection methods are shown in Table 1 for an average gait speed of 1.4 m/s and distances of 80 m and 2.4 m. The table shows that the longer the distance, the more accurate the speed measurement becomes due to the reduction in the weight of the GPS error.

We also conducted tests to evaluate the quality of manual speed measurements using a stopwatch over a distance of 2.4 m as performed by practitioners within medical centres. The results are presented in Section 4.5. We observe a similar order of error between the GPS measurements over straight distances of 80 m and the manual method. The errors are approximately 0.04 m/s and 0.042 m/s, respectively.

However, for short distances (three GPS samples, for two seconds), the error in the GPS measurements is greater (0.098 m/s). Thus, if smartphones are used for gait speed estimation, the distance considered should be greater than that used in indoor tests. We believe that this is not a drawback, as longer distance measurements are more representative of patients’ gait speed when walking naturally.

## 6. Conclusions and Future Work

Given the ageing population in most countries and the associated decrease in economic resources, it is crucial to automate the monitoring or tracking of chronic patients. The research community is making a considerable effort in this regard. Although there are studies that conduct quite complex analyses of the real-life behaviour of frail patients, our short-term goal is to continuously monitor the frailty tests already conducted by health practitioners, with gait speed being one of these tests.

The less costly and, probably, the more accurate way to achieve this goal is to monitor the patient in their daily activities. In this article, we explore the feasibility of complementing indoor gait speed tests conducted by health practitioners with smartphone devices to assess the frailty status of patients. This approach would enable the continuous and comprehensive monitoring of frailty.

Specifically, the errors incurred when measuring the walking speed with the GPS unit of a smartphone have been assessed. These measurements are compared with photo-detection, which is considered the ground truth. The experiments have shown a very high correlation between both methods. For the straight paths of 80 m, we observe a similar level of error to that of the manual indoor tests using a stopwatch, approximately 0.04 m/s of velocity error.

Nevertheless, the daily monitoring of natural walking to detect speed variations presents two main concerns. One is how to summarise quantitative data to characterise the walking speed. For instance, sufficiently long straight segments in which patients walk at a constant speed should be extracted from a monitored walk. Another concern is how to address dynamic external factors, such as topographic variations in the path being walked, weather conditions, or obstacles.

Finally, it is widely acknowledged that real-life monitoring opens up numerous possibilities. In addition to the more representative and continuous monitoring of patients, more data on physical and social activity, or even on the mood of patients, can be collected. These represent our medium-term objectives, while obviously considering the privacy implications of these practices.

## Figures and Tables

**Figure 1 sensors-24-04588-f001:**
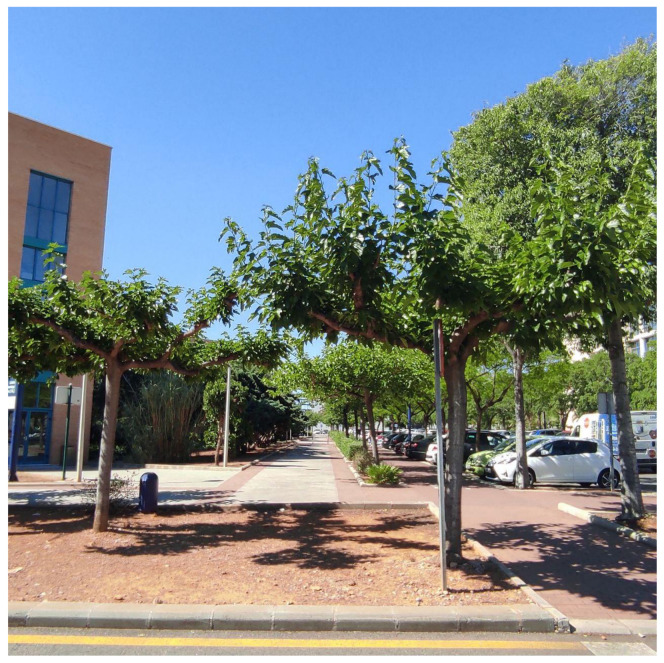
Scenario used in the experiments: a straight, flat path surrounded by trees and buildings.

**Figure 2 sensors-24-04588-f002:**
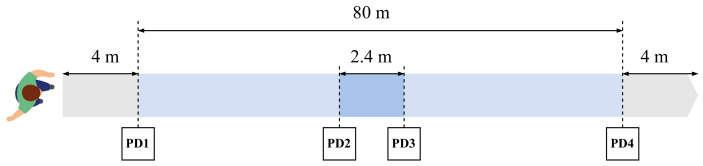
Scenario used in the experiments. Participants start and end walking 4 m before and after the marks of the 80 m walk. Photo-diodes are placed at the extremes of the 80 m (PD1 and PD4) and at 39 and 41.4 m of the walk (PD2 and PD3).

**Figure 3 sensors-24-04588-f003:**
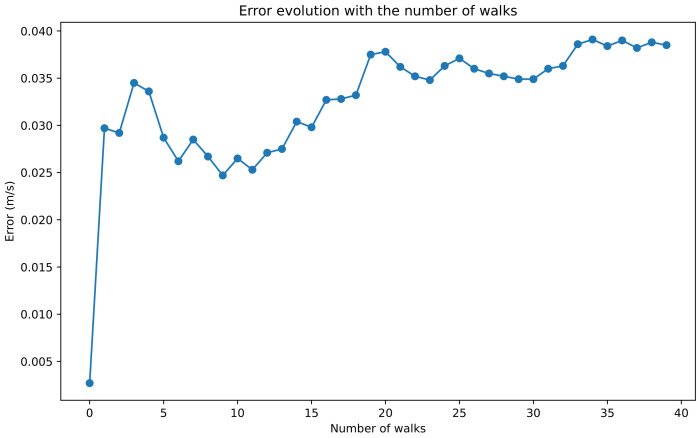
Average absolute error of the measured speed in m/s as function of the number of walks.

**Figure 4 sensors-24-04588-f004:**
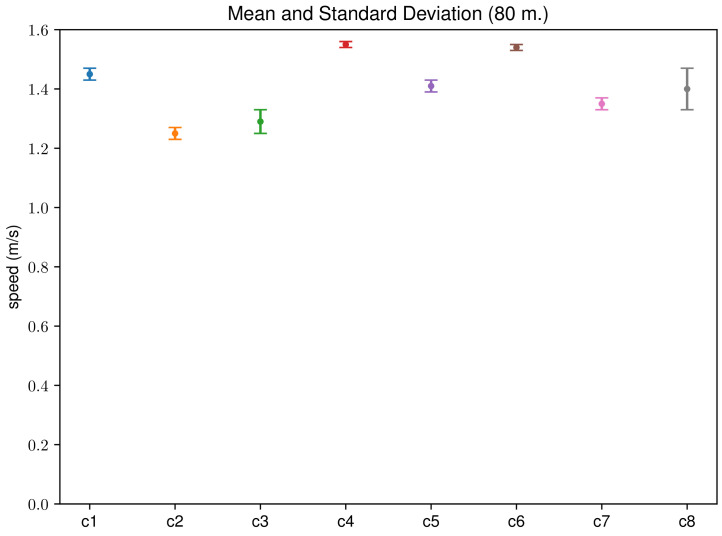
Average and standard deviation summarizing the 5 walks for each participant. Data correspond to the 80 m walk measured by photo-detection.

**Figure 5 sensors-24-04588-f005:**
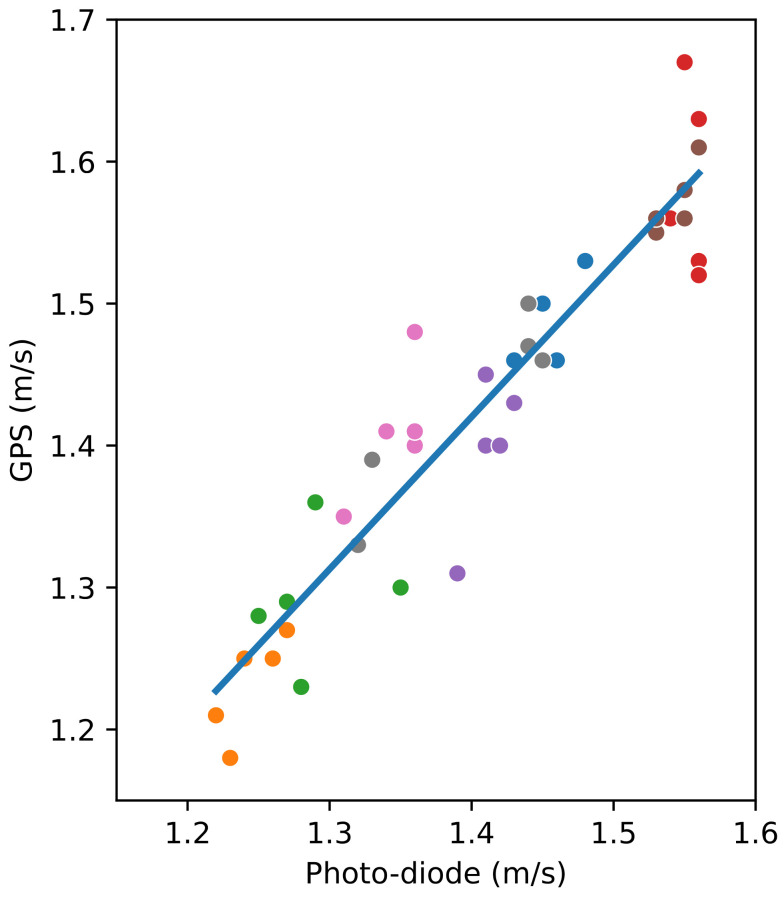
Comparing the speed obtained using GPS and photo-diodes for the 80 m walk. Participants are represented with different colours. The blue line shows the linear regression, with the slope being 1.0723 and the intercept being −0.0850.

**Figure 6 sensors-24-04588-f006:**
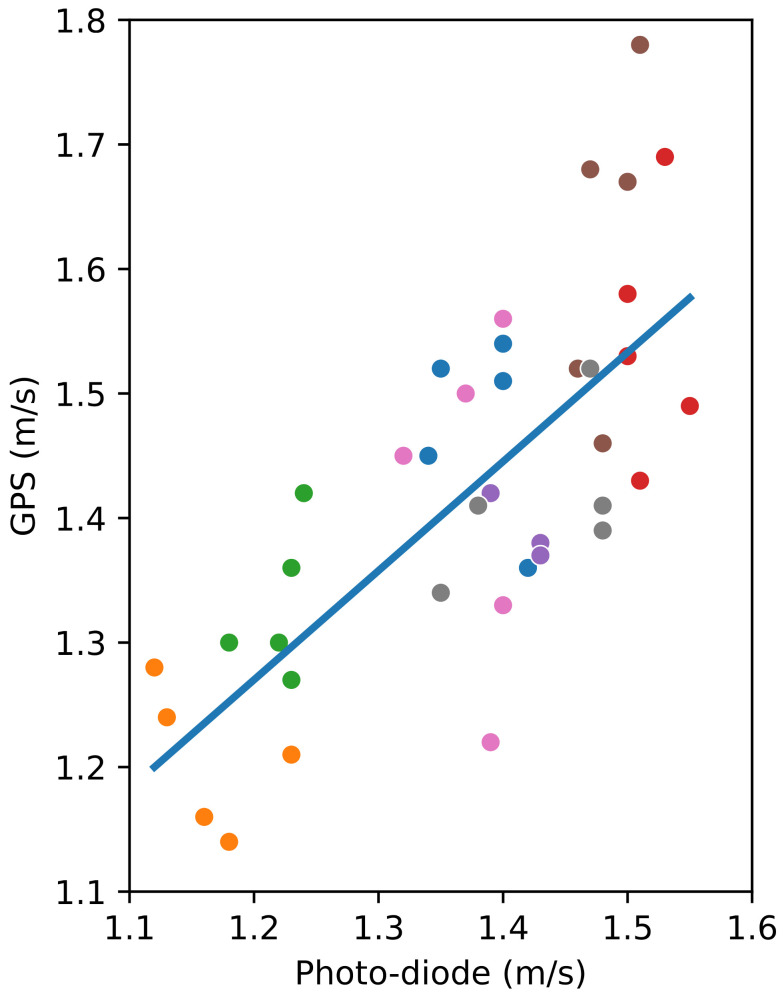
Comparing the speed obtained using the GPS and photo-diodes for a distance of 2.4 m in the central part of the walk. Participants are represented with different colours. The line represents the linear regression, the slope being 0.8768 and the intercept being 0.218.

**Figure 7 sensors-24-04588-f007:**
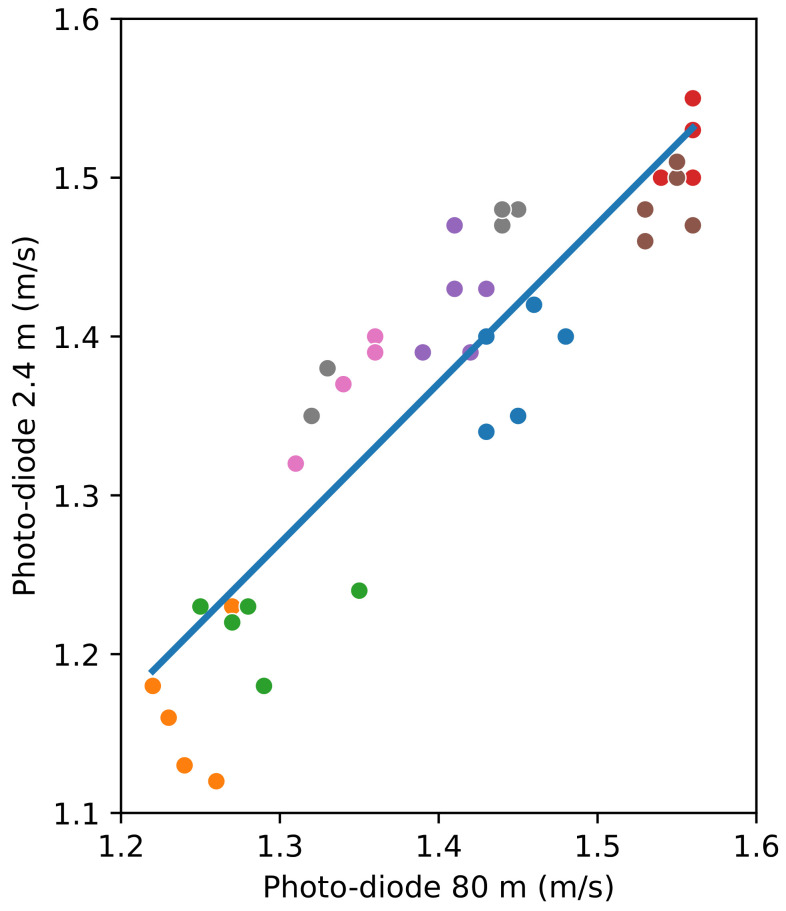
Comparing the speeds measured by photo-detection for distances of 80 m and 2.4 m. Participants are represented with different colours. The line represents the linear regression, the slope being 1.0365 and the intercept being −0.0828.

**Figure 8 sensors-24-04588-f008:**
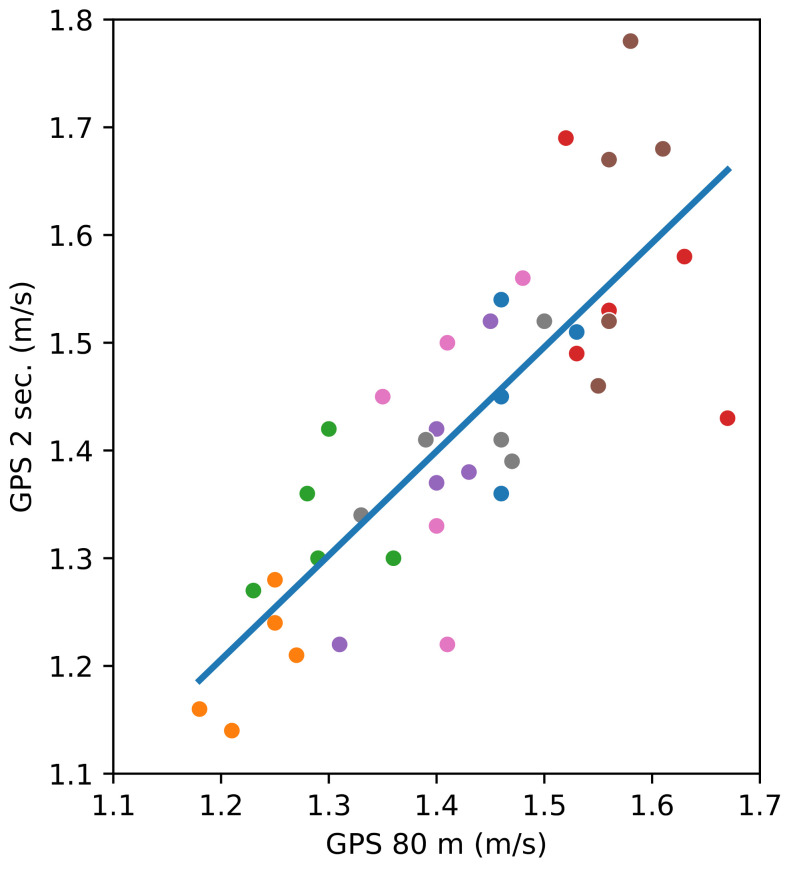
Comparing the speed measured by GPS during the 80 m walk with the speed calculated over a 2 s interval of walking. Participants are represented with different colours. The line represents the linear regression, the slope being 0.967 and the intercept being 0.0457.

**Figure 9 sensors-24-04588-f009:**
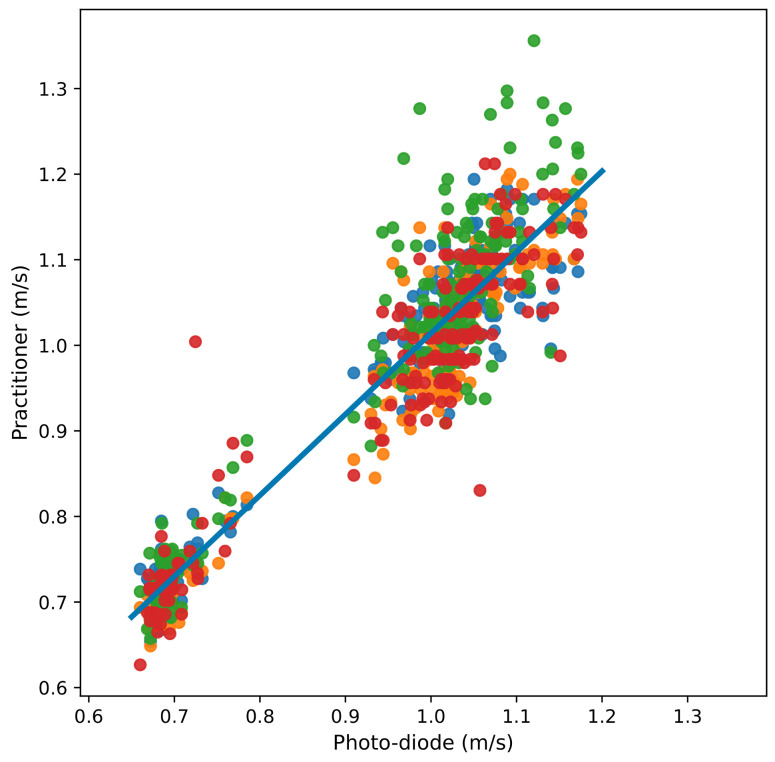
Comparing the speed measured by photo-diodes with the speed measured manually by four practitioners. Practitioners are represented with different colours. The line represents the linear regression, the slope being 0.947 and the intercept being 0.0670.

**Figure 10 sensors-24-04588-f010:**
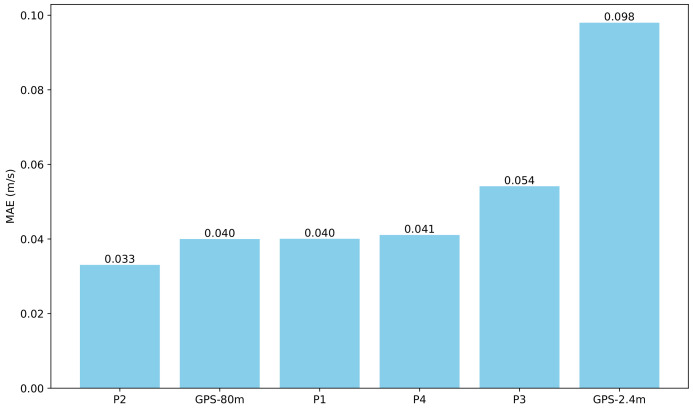
Mean absolute error for the four practitioners and the speed obtained with the GPS (distances of 2.4 m and 80 m).

**Table 1 sensors-24-04588-t001:** Differences observed between gait speed measured with the GPS and by photo-detection (ground truth) in m/s for distances of 80 m and 2.4 m.

Speed Measure Differences (m/s)		
Differences	80 m	2.4 m
Maximum	0.12	0.26
Minimum	0	0
Average	0.04	0.098

## Data Availability

The data presented in this study are available on request from the corresponding author due to ethical reasons.

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
