# Peer review of "GPS Suitability for Physical Frailty Assessment"

_sensors, 2024, doi:10.3390/s24144588_

Round 1

Reviewer 1 Report

Comments and Suggestions for Authors

This paper investigates the feasibility of using smartphones for monitoring the frailty of patients, which has practical application value. However, the article reads more like a research report than a scientific paper. To enhance its scientific rigor, improvements can be considered in the following areas:

  1. The results of smartphone speed measurements are highly correlated with the performance of the phone and how pedestrians hold the phone. Whether these data are consistent across 40 tests or random; if random, there will be differences between different tests. How to deal with these differences.
  2. Due to the influence of various external factors, there are bound to be outliers in the results of smartphones. How to detect and eliminate outliers to ensure the correctness of the results has not been discussed in this paper.

Author Response

Dear Editor,

Please find attached our responses to the reviewer's comments.

Best regards,

Miguel Pérez-Francisco

Reviewer 2 Report

Comments and Suggestions for Authors

Hello, overall I thought this was a well designed and well presented research study.  I think the sample size of 8 participants was pretty low, it would have been nice to see a larger sample and a more varied sample in terms of gait speed, but overall I thought this was very nicely done and an important study.

Specific section comments:

Abstract - some difficult sentence structures that are hard to understand as presented.  for example, in line 4 - "...it is compared the gait speed measured outdoors using smartphones..." is quite awkwardly phrased.  I think the manuscript could benefit from an editorial review by someone whose first language is English if you have it available.  

Section 2, line 57 - I think the "or" should be "of" - unless I am not understanding the sentence here, I think this is likely just a typographical error, but if "or" is correct, then consider re-working the sentence to enhance understandability.

General comment about your findings - did the error measures have any dependency on gait speed?  (i.e. were there bigger or smaller errors associated with faster or slower speeds).  It was hard to tell from the data and perhaps there was not enough variability in the sample speeds to determine this, but it would be interesting to note.  The sample used for data collection all seemed rather young (24 - 60 years old), and I suspect none had very slow gait speeds, so I was just wondering if there is any impact of the actual gait speed on the size of the errors as this would be an important issue for the intended population.

Comments on the Quality of English Language

There are some phrases used that are not very standard in English language manuscripts (such as use of the phrase "ground truth" where we might use "gold standard") - but I did not find the language to be a barrier to understanding the meaning of the article.  I do think it would be worth an English language editorial review, but it is basically understandable as written. Some of the sentence structures are difficult to follow, but again, this does not severely impede the ability to understand the message.

Author Response

(The authors gave the same response as above.)

Round 2

Reviewer 1 Report

Comments and Suggestions for Authors

The author has provided the necessary responses and has no further questions.